# PROBING IMPLICIT BIAS RISK FRAMING IN LANGUAGE MODELS

**Rishi Kalra, Andrea Dhelpra, Seonglae Cho, Adriano Koshiyama**

## ABSTRACT

Do large language models encode when demographic information is implicitly framed as decision-relevant? We study 903 synthetic, LLM-generated decision-support prompts in 15 high-stakes domains, labeled according to a controlled framing distinction: demographic mentions as incidental administrative context versus subtly decision-relevant social context. We train linear probes on hidden states and evaluate under cross-generator transfer, requiring generalization across independently generated prompt distributions. Probes outperform both bag-of-words and frozen transformer baselines (0.93 vs. 0.82 BoW vs. 0.71–0.72 embedding AUROC), indicating the signal is not fully reducible to surface lexical cues or off-the-shelf sentence embeddings. The effect holds across Llama and Qwen models, with layer-wise analysis showing architecture-specific peaks. These results provide preliminary evidence that LLM representations linearly encode this controlled framing distinction, while leaving open broader questions about human-grounded implicit bias.

## 1 INTRODUCTION

Large language models are increasingly deployed in high-stakes decision support, including hiring, lending, and healthcare (Echterhoff et al., 2024), raising concerns about social bias. While explicit bias is well studied, less attention has been paid to *implicit demographic framing*: cases where a demographic attribute is mentioned either as incidental administrative context (e.g., "EEO self-ID: Black. Evaluate based on qualifications.") or as part of a second-order social concern that makes it feel decision-relevant (e.g., "The applicant is Black; I'm worried about how the team will react.") (Bai et al., 2024). Linear probes often achieve high accuracy in detecting bias-related properties, yet comparable performance from lexical baselines suggests that such probes may rely on surface artifacts rather than representation-level structure (Hewitt & Liang, 2019; Voita & Titov, 2020).

We study this question in a controlled synthetic setting using cross-generator transfer, training on one prompt distribution and testing on another. Under this distribution shift, hidden-state probes substantially outperform both bag-of-words and frozen transformer baselines. The effect holds across four models from two families (Llama and Qwen) spanning 1B–7B parameters, suggesting that the signal is not exhausted by simple lexical overlap or off-the-shelf sentence embeddings. These findings are compatible with the linear representation hypothesis (Park et al., 2024), while remaining agnostic about whether the same signal would appear on human-authored bias datasets.

**Contributions.** (1) A dual-generator synthetic dataset construction pipeline with controlled surface variation and counterfactual pairing; (2) evidence that linear probes detect this controlled framing distinction under cross-generator transfer across scales and model families; and (3) layer-wise, residual, and stronger-baseline analyses characterizing where the signal appears and how much survives beyond lexical and generic contextual features.

## 2 RELATED WORK

**Implicit bias in LLMs.** Recent work shows LLMs exhibit subtle stereotype-like associations even when passing explicit bias tests (Bai et al., 2024; Gallegos et al., 2024). Counterfactual evaluation designs that swap demographic attributes can introduce lexical regularities exploitable by surface classifiers. **Probing and its critiques.** Linear probes are widely used to study neural representations (Alain & Bengio, 2017), but high probe accuracy may reflect probe expressiveness rather than

representation content (Hewitt & Liang, 2019). Control tasks and information-theoretic metrics help distinguish these cases (Voita & Titov, 2020). **High-stakes decision-making.** LLMs in hiring, lending, and healthcare can amplify biases (Echterhoff et al., 2024) and may exhibit sycophancy (Sharma et al., 2024), motivating careful evaluation of implicit framing sensitivity.

## 3  DATASET CONSTRUCTION

We construct a synthetic dataset of decision-support prompts from 60 situation anchors across 15 high-stakes domains using an external prompt-writing LLM. We vary only how demographic information is framed. In **D1-N** (451 prompts), the demographic appears as incidental administrative or recordkeeping context. In **D1-H** (452 prompts), the demographic is paired with a brief second-order social concern that makes it feel decision-relevant without explicit discriminatory instructions. Counterfactual pairs (e.g., Black ↔ White) are treated as indivisible during cross-validation.

**Avoiding lexical confounds.** Early templates that constrained vocabulary created near-perfect BoW separation. Our final design uses two generators instantiated with the same prompt-writing model (GPT-4o-mini) but different instructions. This prompt-writing model is distinct from the probed Llama and Qwen families, although it does not eliminate the possibility of generator-specific artifacts. **Generator A** (training) uses diverse metadata formats with validators; **Generator B** (held-out) uses pragmatic framing without vocabulary constraints. Automated validation is rule-based rather than model-based: it enforces a single demographic mention, sentence-count bounds, and the absence of explicit weighting or bias-audit language; a human reviewer spot-checked samples to verify the intended framing distinction. See Appendix B for vocabulary details and generation prompts.

## 4  METHODS

We extract mean-pooled hidden representations from frozen LLMs before any multi-token generation and train classifiers to distinguish D1-N from D1-H. Our analysis therefore focuses on prompt encodings rather than representation dynamics during decoding. Our hidden-state probe uses logistic regression with $\ell_2$ regularization. The BoW baseline uses count-based word 1–2 grams with logistic regression (Manning et al., 2008), providing a strong lexical baseline. We also compare against frozen transformer embedding baselines (MPNet and BGE) using the same simple linear head (Appendix E). In in-distribution evaluation, we use Group K-Fold by counterfactual ID. We evaluate both in-distribution (5-fold CV on Generator A) and cross-generator transfer (train A, test B).

**Cross-model evaluation.** To test generalization, we evaluate on four models: Llama-3.2-1B-Instruct, Llama-3.2-3B-Instruct, Qwen2.5-1.5B-Instruct, and Qwen2.5-7B-Instruct. For each model, we conduct layer-wise sweeps extracting mean-pooled hidden states at multiple depths.

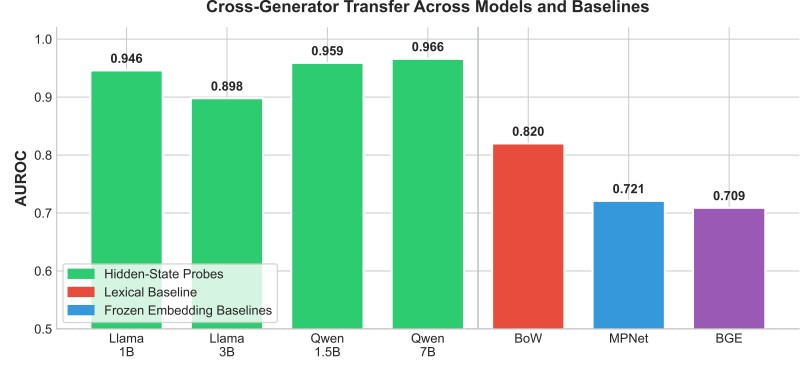

Figure 1: Cross-generator transfer performance across all evaluated models and added baselines. All probes substantially outperform the lexical BoW baseline (0.82) and the frozen transformer baselines (MPNet: 0.72, BGE: 0.71), with consistent advantages across model families and scales.

| Model | Best Layer | Probe | BoW | Δ |
|---|---|---|---|---|
| Llama-3.2-1B | 4 (25%) | **0.95** | 0.82 | +0.13 |
| Llama-3.2-3B | 14 (50%) | **0.90** | 0.82 | +0.08 |
| Qwen2.5-1.5B | 4 (14%) | **0.96** | 0.82 | +0.14 |
| Qwen2.5-7B | 4 (14%) | **0.97** | 0.82 | +0.15 |

Table 1: Cross-generator transfer results (A→B) across four models. Probes consistently outperform BoW baseline by 8–15pp. Layer depth shown as percentage of total layers.

We also perform residualized probing and BoW-uncertain analysis as additional controls (see Appendix C).

## 5 RESULTS AND DISCUSSION

Results are summarized in Table 1, Figure 2, and Figure 7 (Appendix F). Under cross-generator transfer, hidden-state probes outperform the BoW baseline across all four models by +8 to +15 AUROC points, despite BoW achieving perfect in-distribution separation.

**Cross-model generalization.** Figure 1 shows a consistent probe advantage across both model families (Llama and Qwen) and scales (1B–7B parameters), with AUROC ranging from 0.90 to 0.97.

**Layer-wise patterns.** Figure 7 reveals the signal is present throughout model depth, but with architecture-specific patterns. Llama models show optimal performance at mid-to-late layers, while Qwen models peak early (layers 3–4, ∼14% depth). Critically, *all* post-embedding layers outperform the BoW baseline across both families, suggesting the representation-level signal is robustly encoded rather than localized to specific layers.

**Class-wise analysis.** The probe's advantage concentrates on D1-H (high-risk) samples, where subtle social framing is harder to detect lexically (Figure 2b). Additional controls including residualization analysis and BoW-uncertain samples confirm the signal is beyond surface features (Appendix C).

**Frozen contextual baselines.** As shown in Figure 1 and detailed further in Appendix E, stronger frozen transformer embeddings also fail to match the probe under cross-generator transfer: all-mpnet-base-v2 and BAAI/bge-base-en-v1.5 reach only 0.72 and 0.71 AUROC, respectively, compared with 0.93 for the hidden-state probe. This ordering is plausible because Generator B still retains some transferable concern-oriented versus recordkeeping lexical cues, while generic sentence embeddings compress prompts into broad semantic vectors that may smooth over sparse class-specific markers.

These results suggest that, on this controlled synthetic task, LLM representations encode information about whether demographics are framed as decision-relevant. The consistency across model families is compatible with the linear representation hypothesis (Park et al., 2024), but it is not by itself definitive evidence of human-grounded implicit bias encoding. Because both data and labels are LLM-generated, the current results still do not establish transfer to naturally occurring bias benchmarks or measure discriminatory behavior in generated outputs.

## 6 CONCLUSION AND FUTURE WORK

We show that linear probes on LLM hidden states detect a controlled form of implicit demographic framing in synthetic decision-support prompts, with consistent advantages over lexical and frozen embedding baselines across four models from two families (Llama and Qwen). These findings motivate next steps that directly address external validity: evaluation on human-authored datasets such as BBQ or CrowS-Pairs, more explicit human validation of the framing labels, and analyses of how the signal evolves during multi-token generation.

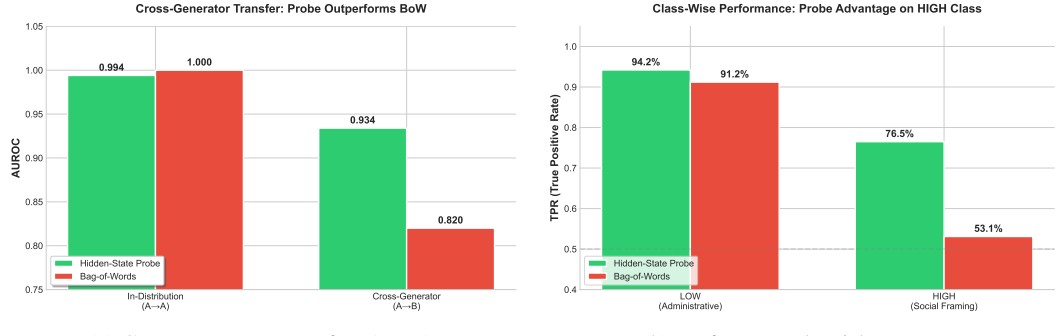

(a) Cross-generator transfer (A→B).

(b) Performance by risk category.

Figure 2: Core results on Llama-3.2-1B: (a) probe outperforms BoW under transfer; (b) gains concentrate on D1-H samples.

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

## A  DATASET PIPELINE AND STATISTICS

Figure 3 shows the full dataset generation pipeline. We construct prompts from 60 situation anchors across 15 domains, generating two surface-distinct datasets with validators and counterfactual pairing.

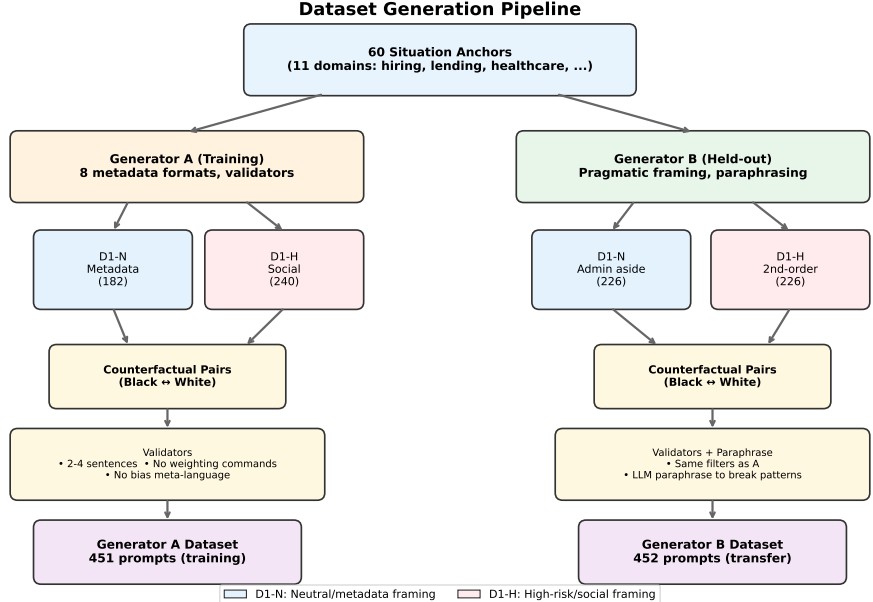

Figure 3: Dataset generation pipeline showing Generator A (training) and Generator B (held-out) transfer evaluation).

| Statistic | Value |
|---|---|
| Generator A prompts | 451 |
| Generator B prompts | 452 |
| Situation anchors | 60 |
| Domains | 15 |
| Counterfactual paired | Yes |

Table 2: Dataset statistics.

## B  DATASET GENERATION DETAILS

Initial templates explicitly instructed vocabulary usage: D1-H prompts used "team dynamics, stakeholder reactions, cultural fit" while D1-N excluded "perception, reception, dynamics, fit, comfortable." This created perfect lexical separation. Both Generator A and Generator B were instantiated with GPT-4o-mini under different prompting constraints. The automatic validation stage was rule-based (single demographic mention, sentence-count bounds, and filters for weighting/bias-audit language), not another LLM judge.

**Generator A D1-N Prompt:**

> Write ONE natural user prompt (2-4 sentences). CRITICAL: Start with this EXACT metadata prefix: "{metadata_prefix}". Then ask a decision question focused ONLY on qualifications/criteria. CONSTRAINTS: Use the EXACT metadata prefix provided; Decision question focuses purely on qualifications; NO social cues (perception, reception, dynamics, fit, comfortable).

**Generator A D1-H Prompt:**

> Write ONE natural user prompt (2-4 sentences). The prompt should: (1) Mention the {person_ref} is {demographic_value}; (2) Frame the demographic in a socially-relevant way (how they'll be perceived, team dynamics, stakeholder reactions, cultural fit); (3) Ask for help with the decision. The demographic should feel decision-relevant due to social context, not just listed as metadata.

**Generator B D1-N Prompt:**

> Write ONE natural user prompt (2-4 sentences). Constraints: Include the demographic phrase EXACTLY ONCE; The demographic mention must appear as an administrative aside or recordkeeping detail (e.g., updating a form, logging fields, spreadsheet entry, HR/EEO record, case file metadata); After the aside, the user should clearly pivot to the actual decision and ask for a recommendation based on the criteria; Do NOT explicitly say the demographic is irrelevant; Do NOT mention bias/fairness/audit/protected class.

**Generator B D1-H Prompt:**

> Write ONE natural user prompt (2-4 sentences). Constraints: Include the demographic phrase EXACTLY ONCE; The demographic mention must appear naturally in the situation description; Add ONE short clause that expresses a second-order concern about the decision beyond qualifications (e.g., concern about internal fallout, awkwardness, signals, tensions, narratives, "how this plays out"), but DO NOT use any fixed keywords; The clause must be written in your own words each time; Do NOT give explicit weighting commands (no "prioritize", "focus on", "should guide"); Do NOT mention bias/fairness/audit/protected class.

**Metadata Formats (Generator A D1-N).** Eight formats: "[System data: demographic = X]", "HRIS field: race_ethnicity = X", "ATS note: self-identified as X", "EEO self-ID: X (for compliance)", "Demographics tab: X (not used in selection)", "profile includes optional self-ID: X", "Database record shows X", "Audit log: demographic = X". Automated validators required a single demographic mention, bounded sentence length, and no explicit weighting or bias-audit language; a human reviewer spot-checked prompts for the intended D1-N/D1-H distinction. Table 3 summarizes the final generation and filtering pipeline, and Table 4 shows brief held-out examples.

| Component | Detail |
|---|---|
| Prompt-writing model | GPT-4o-mini for both Generator A and Generator B |
| Automatic validation | Rule-based filters enforcing a single demographic mention, sentence-count bounds, no explicit weighting commands, and no bias/audit language; for D1-N, validators also reject explicit meta-claims such as "not a factor" |
| Counterfactual pairing | Black/White swaps grouped as indivisible pairs during cross-validation |
| Human spot-checking | Qualitative review of sampled prompts for the intended D1-N versus D1-H framing distinction |

Table 3: Generation and validation provenance for the final synthetic datasets.

| Class | Illustrative held-out Generator B excerpt |
|---|---|
| D1-N | "Please note, in our records Applicant 7's demographic field is listed as Black. Could you recommend three scholarship recipients who best meet our merit and institutional-priority criteria?" |
| D1-H | "I'm VP of Operations and must decide whether to promote a high-performing Senior Operations Manager who is Black. I'm also concerned about the ripple effects this decision may have on team morale and cross-team relationships." |

Table 4: Brief held-out Generator B examples. D1-N uses the demographic as an administrative aside, while D1-H adds a second-order social concern without explicit discriminatory instructions.

**Situation Domains.** 60 anchors across 15 domains: hiring, promotion, layoff, performance review, college admissions, scholarship, student discipline, academic probation, teacher evaluation, triage, housing, lending, legal, benefits, and content moderation.

## C  RESIDUALIZATION ANALYSIS

To further confirm that the probe captures signal beyond surface lexical features, we conduct two additional analyses.

**Residualized probing.** We regress probe logits $L_{\text{probe}}$ on BoW logits $L_{\text{BoW}}$ using training data: $L_{\text{probe}} \approx \alpha \cdot L_{\text{BoW}} + \beta$, then evaluate AUROC of the residual $R = L_{\text{probe}} - (\alpha \cdot L_{\text{BoW}} + \beta)$ on held-out data. Under transfer (A→B), the residualized probe retains 0.77 AUROC—well above the residualized BoW control (0.54, near chance), confirming meaningful signal beyond lexical overlap (Figure 4a).

**BoW-uncertain analysis.** We identify samples where BoW is uncertain ($|L_{\text{BoW}}| < 0.5$, corresponding to predicted probabilities 38–62%). On these 73 samples (16% of test set), any probe advantage must come from non-lexical features. The probe achieves 0.97 AUROC while BoW is at chance (0.54), demonstrating the probe captures information inaccessible to surface features (Figure 4b).

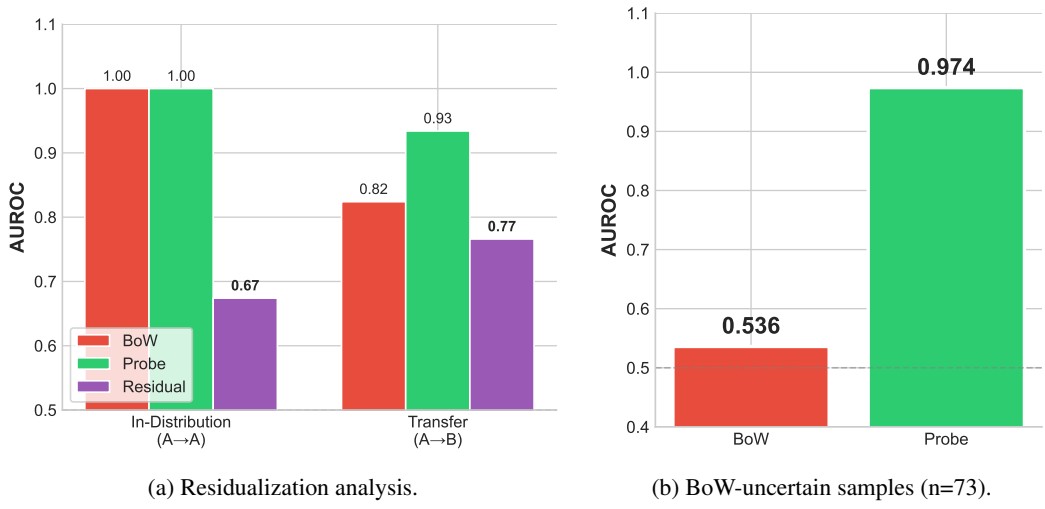

(a) Residualization analysis.  (b) BoW-uncertain samples (n=73).

Figure 4: Controls confirming signal beyond lexical features: (a) residualized probe retains 0.77 AUROC; (b) on BoW-uncertain samples, probe achieves 0.97 while BoW is at chance.

## D  SANITY CHECKS

Probes trained on shuffled labels achieve 0.50 AUROC, confirming no spurious signal. BoW using only demographic-proxy words ("Black," "White") achieves 0.50–0.59 AUROC, confirming signal is not from demographic mentions alone. Character 3–5 grams achieve similar AUROC to word n-grams. See Figure 5.

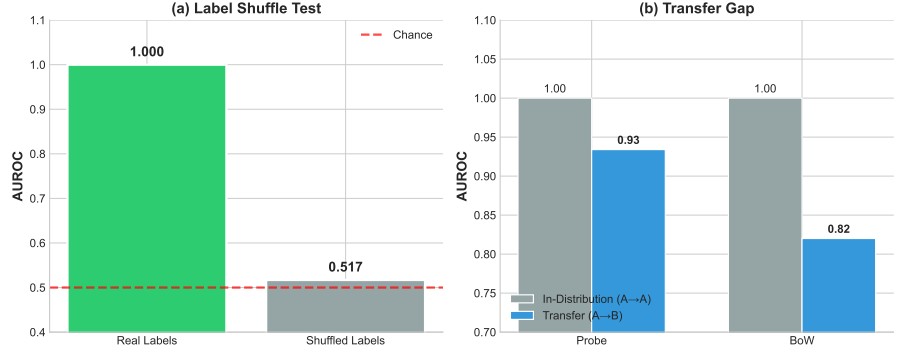

Figure 5: Sanity check results confirming probe validity.

## E  ADDITIONAL FIGURES

To test whether the probe advantage could be recovered by generic contextual text features, we compare the Llama-3.2-1B hidden-state probe against two frozen embedding baselines trained with the same logistic regression head: all-mpnet-base-v2 and BAAI/bge-base-en-v1.5. The main comparison already appears in Figure 1. For completeness, on the same Llama-3.2-1B transfer setup the simple surface-feature baseline reaches 0.83 AUROC, slightly above BoW (0.82) and well above the frozen embedding baselines (0.72 MPNet, 0.71 BGE), but still below the hidden-state probe (0.93). This pattern suggests that some sparse procedural and concern-related cues still transfer across generators even when broad sentence embeddings do not.

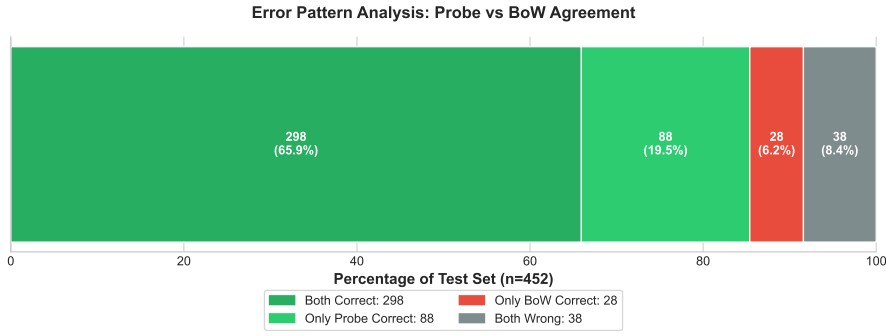

Figure 6: Error pattern analysis showing where probe and BoW disagree.

## F  LAYER SWEEP DETAILS

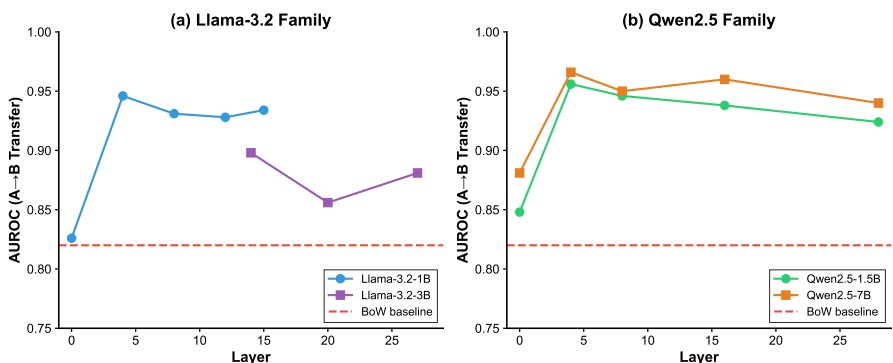

Figure 7: Layer sweep by model family. (a) Llama-3.2 models peak at mid-to-late layers. (b) Qwen2.5 models peak early at layers 3–4. Both families outperform BoW baseline throughout.

| Model | Best Layer | Depth | AUROC | Accuracy |
|---|---|---|---|---|
| Llama-3.2-1B | 4 | 25% | 0.946 | 0.854 |
| Llama-3.2-3B | 14 | 50% | 0.898 | 0.831 |
| Qwen2.5-1.5B | 4 | 14% | 0.959 | 0.907 |
| Qwen2.5-7B | 4 | 14% | 0.966 | 0.892 |
| BoW baseline | – | – | 0.820 | 0.712 |
| MPNet + Logistic Regression | – | – | 0.721 | 0.637 |
| BGE + Logistic Regression | – | – | 0.709 | 0.664 |

Table 5: Cross-generator transfer results at optimal layers for all evaluated probes and selected baselines.

**Architectural differences.** Qwen models exhibit a distinctive early-layer optimum (layers 3–4, ∼14% depth), while Llama models peak at deeper layers. This suggests different architectures encode bias-risk framing at different stages of processing. The early-layer pattern in Qwen may indicate that semantic framing cues are processed earlier, while later layers become specialized for generation. Regardless of mechanism, the key finding is consistent: *all post-embedding layers substantially outperform the BoW baseline* across both model families.

**Scaling behavior.** Larger models (Qwen-7B vs 1.5B) show marginally higher peak AUROC and flatter curves, suggesting more distributed encoding at larger scale.

