# OpenReview forum: "Probing Implicit Bias Risk Framing in Language Models"
_ICLR.cc/2026/Workshop/AFAA — AFAA 2026 Poster_

### Official Review · Reviewer_wtjT · 2026-02-14
**A well-motivated and methodologically short paper introducing a promising probing framework for implicit demographic framing, which would benefit from clearer construct grounding and validation beyond fully synthetic data.**

**Rating:** 4
**Confidence:** 5

**Summary:**

This paper investigates whether large language models (LLMs) encode information about implicit demographic framing in decision-support prompts. The authors construct a synthetic dataset of 903 prompts across 15 high-stakes domains. Using linear probes, the authors show that LLM probing performance substantially outperforms a TF-IDF bag-of-words baseline under distribution shift. The authors interpret these findings as evidence that LLM internal representations linearly encode demographic framing information.

**Strengths:**

The paper is well-positioned for the workshop and introduces an interesting topic: implicit biases in high-stakes domains.

The dual-generator dataset construction is a notable strength, as it attempts to isolate implicit demographic framing effects by creating lexical separation around implicit clues on cultural fit, team dynamics, perceptions, and reactions. The dataset does not rely on explicit bias keywords that could trivialize the task.

The paper introduces a robust probing analysis method studying LLMs' internal representations under distribution shifts. This is a conscientious attempt to address a known critique of probing that might rely on lexical artifacts and superficial statistical clues instead of capturing notions of bias. The paper clearly emphasizes this phenomenon, showing that TF-IDF representation probing achieves near-perfect performance for in-distribution classification, showing a performance gap only under distribution shifts.

The paper introduces multiple control strategies (random shuffling and transfer gaps) to strengthen the claim.

The paper studies multiple models with various architectures and applies the probing approach to multiple layers, reinforcing the robustness of their approach.

**Weaknesses:**

The following suggestions are aimed at clarifying and strengthening the contribution to a workshop audience.
- The paper does not clearly define or theoretically ground the concept of “implicit demographic framing.” The distinction between D1-H and D1-N appears to be operationalized via the generator instructions rather than a principled definition of implicit bias, raising concerns about construct validity. Providing a short, explicit definition and 1–2 illustrative examples in the main text would help readers better assess the framing distinction between the D1-H and D1-N cases and assess their relevance to implicit bias. Finally, the control results produced by the human reviewer and the criteria they used to validate the examples are missing, making it impossible to confirm that the examples are qualitative.
- The dataset is fully LLM-generated, and the paper does not disclose which model was used to generate it. The problem here, when comparing probes from LLMs and TF-IDF on LLM-generated cases, is that the results don’t necessarily reveal implicit bias encoding. It may show representational compatibility between LLMs. Probes trained on representations from similar models may detect generator artifacts. These might be easily recoverable using transformer features, but not with TF-IDF, without introducing bias per se. Therefore, I agree with the authors that it will be necessary to compare their approach on a human-based dataset, such as CrowS-Pairs or the BBQ dataset, to at least have a baseline in that regard.

---

### Official Review · Reviewer_4DcA · 2026-02-20
**review of "Probing Implicit Bias Risk Framing in Language Models"**

**Rating:** 4
**Confidence:** 4

**Summary:**

This paper investigates if models can internally detect when demographic information is being framed as a relevant factor for making high-stakes decisions. By analyzing the hidden layers of models like Llama and Qwen. This paper demonstrate that linear probes on hidden states can classify implicit demographic framing.

**Strengths:**

- The research addresses a highly relevant and interesting question regarding how LLMs internally process demographic information when subtly framed as a decision factor.
- The linear probing methodology is technically sound and offers a versatile approach for future tasks like model steering, debiasing, and automated detection.
- The generated dataset contributes to the field of mechanistic interpretability by providing a resource to study how abstract concepts are encoded in model representations.

**Weaknesses:**

- The conclusions are not fully supported by the current evidence; the paper needs more experimental analysis. The scope and depth of the experimental analysis are well-suited for the requirements of a tiny track submission but not for supporting the claim of the paper.
- The study focuses only on hidden states during next-token prediction; analyzing how these states evolve over the course of generating many tokens is necessary for a complete understanding.
- To prove generalization, the authors should test the probe's transferability on other datasets, such as those containing explicit bias, to see if it performs better than a random baseline.
- The paper lacks qualitative depth and would benefit from including specific examples of the generated prompts to clarify the difference between incidental and risk-based framing.

---

### Meta-Review · Area_Chair_Ky6s · 2026-02-21

**Recommendation:** Tiny/Short Papers Track
**Confidence:** 5

**Metareview:**

Reviewers agree that the short paper is sound and well-motivated with clear potential impact. The proposed probing technique is well appreciated. They also suggested ways for improvement for the paper both in terms of clarity and richness of the experiments. I believe the paper can benefit from discussions with the community and can be of interest to the workshop attendees.

---

### Decision · Program_Chairs · 2026-03-02

Accept (Poster)